# Increased therapeutic effect on medullary thyroid cancer using a combination of radiation and tyrosine kinase inhibitors

Viktor Sandblom[1]*, Johan Spetz[1], Emman Shubbar[1], Mikael Montelius[1], Ingun Ståhl[1,2], John Swanpalmer[1,2], Ola Nilsson[3], Eva Forssell-Aronsson[1,2]

**1** Department of Radiation Physics, Institute of Clinical Sciences, Sahlgrenska Cancer Center, Sahlgrenska Academy, University of Gothenburg, Gothenburg, Sweden, **2** Department of Medical Physics and Biomedical Engineering, Sahlgrenska University Hospital, Gothenburg, Sweden, **3** Department of Pathology, Institute of Biomedicine, Sahlgrenska Cancer Center, Sahlgrenska Academy, University of Gothenburg, Gothenburg, Sweden

* viktor.sandblom@vgregion.se

**Data Availability Statement:** All relevant data are within the manuscript.

## Abstract

Since patients with medullary thyroid cancer (MTC) often have metastatic disease at the time of diagnosis, the development of efficient systemic treatment options for MTC is important. Vandetanib and cabozantinib are two tyrosine kinase inhibitors (TKIs) that were recently approved by FDA and EMA for systemic treatment of metastatic MTC. Additionally, since MTC is of a neuroendocrine tumour type, treatment with radiolabelled somatostatin analogues (*e.g.* $^{177}$Lu-octreotate) is a valid option for patients with MTC. The aim of this study was to investigate the potentially increased therapeutic effect of combining radiation therapy with these TKIs for treatment of MTC in a mouse model. Nude mice carrying patient-derived MTC tumours (GOT2) were treated with external beam radiotherapy (EBRT) and/or one of the two TKIs vandetanib or cabozantinib. The tumour volume was determined and compared with that of mock-treated controls. The treatment doses were chosen to give a moderate effect as monotherapy to be able to detect any increased therapeutic effect from the combination therapy. At the end of follow-up, tumours were processed for immunohistochemical (IHC) analyses. The animals in the combination therapy groups showed the largest reduction in tumour volume and the longest time to tumour progression. Two weeks after start of treatment, the tumour volume for these mice was reduced by about 70–75% compared with controls. Furthermore, also EBRT and TKI monotherapy resulted in a clear anti-tumour effect with a reduced tumour growth compared with controls. The results show that an increased therapeutic effect could be achieved when irradiation is combined with TKIs for treatment of MTC. Future studies should evaluate the potential of using $^{177}$Lu-octreotate therapy in combination with TKIs in patients.

## Introduction

Medullary thyroid cancer (MTC) is a very rare disease, much less common than other types of thyroid cancer, *e.g.* papillary and follicular thyroid cancer [1]. In contrast to these other forms

**Funding:** This study was supported by grants from the Swedish Research Council (grant no. 21073), the Swedish Cancer Society (grant no. 3427), BioCARE – a National Strategic Research Program at the University of Gothenburg, the Swedish state under the agreement between the Swedish government and the county councils – the ALF-agreement (ALFGBG-725031), the King Gustav V Jubilee Clinic Cancer Research Foundation to EFA, as well as grants from the Sahlgrenska University Hospital Research Funds, the Assar Gabrielsson Cancer Research Foundation, the Adlerbertska Research Foundation, the Herbert & Karin Jacobsson Foundation, the Royal Society of Arts and Sciences in Gothenburg (KVVS), and the Wilhelm and Martina Lundgren Research Foundation to VS. The funders had no role in study design, data collection and analysis, decision to publish, or preparation of the manuscript.

**Competing interests:** The authors have declared that no competing interests exist.

of thyroid cancer, MTC originates in the parafollicular C-cells of the thyroid, resulting in a characteristic hormone producing feature. Levels of the hormone calcitonin are often elevated in patients with MTC and can therefore be used as a biomarker for the disease. MTC can occur either sporadically (about 75%) or as a hereditary form. The latter is often part of multiple endocrine neoplasia type 2 (MEN2) syndrome, frequently caused by mutations in the *RET* proto-oncogene. The *RET* proto-oncogene is mutated in most hereditary and about 50% of sporadic MTC and can serve as a molecular target for anti-cancer drugs [2–4]. As for many other types of cancer, the prognosis for MTC patients depends on the extent of metastatic spread. For patients diagnosed with local disease, the 10-year survival rate is above 95%, compared with about 75% and 40% for patients with regional and distant metastases, respectively [5]. Since about half of all MTC patients have metastatic spread at the time of diagnosis, better systemic therapy regimens are warranted.

Tyrosine kinase inhibitors (TKIs) offer a new and promising treatment option for patients with metastatic MTC by targeting the *RET* proto-oncogene frequently mutated in MTC. Two TKIs that have shown promising results in clinical phase I-III studies are vandetanib and cabozantinib [6–10]. On the basis of these results, the U.S. Food and Drug Administration (FDA) and the European Medicines Agency (EMA) has approved both vandetanib and cabozantinib for treatment of patients with progressive locally advanced and/or metastatic MTC. Furthermore, in the latest guidelines from the American Thyroid Association, vandetanib and cabozantinib are both strongly recommended for single-agent first-line therapy in patients with advanced progressive MTC [1].

In addition to *RET* mutations, overexpression of somatostatin receptors (SSTRs) is common in MTC [11, 12]. This overexpression enables treatment with radiolabelled somatostatin analogues such as $^{177}$Lu-octreotate or $^{90}$Y-octreotide–a treatment method included in the concept peptide receptor radionuclide therapy (PRRT). Since its introduction during the 1990s, PRRT has been successfully used for many cancers overexpressing SSTRs, including MTC and other neuroendocrine tumours (NETs) [13–17]. Furthermore, $^{177}$Lu-octreotate was recently approved by FDA and EMA for treatment of gastroenteropancreatic NETs (GEP-NETs). However, healthy organs, such as the kidneys and bone marrow, restrict the amount of drug that can be safely administered to a patient. The treatment protocol using $^{177}$Lu-octreotate states the maximum administered activity and the number of treatment cycles, which results in low frequency of side effects, but also undertreatment of most patients. New treatment strategies are required to increase the cure rate after this treatment. One option for optimisation could be to administer PRRT in combination with another drug, *e.g.* a TKI such as vandetanib or cabozantinib, in order to increase the effect on tumour tissue [18]. To the authors' knowledge, the combination of ionising radiation and vandetanib or cabozantinib has not previously been studied in MTC.

The aim of this study was to investigate the potentially increased therapeutic effect of combining radiation therapy with vandetanib or cabozantinib for treatment of human MTC in a mouse model. Due to reasons further elaborated on below, irradiation was done externally, while the intention for clinical use in the future will be systemic PRRT combined with TKIs.

## Material and methods

In this study, nude mice transplanted with patient-derived MTC were treated by external beam radiotherapy (EBRT) and/or one of the two TKIs vandetanib and cabozantinib. The tumour volume was followed and compared with that in mock-treated mice. All experiments were approved by the Ethical Committee on Animal Experiments in Gothenburg, Sweden.

## Animal model

GOT2 are patient-derived MTC cells originally established in Gothenburg [19]. Briefly, tumour tissue was collected from a patient with a sporadic *RET*-driven MTC and transplanted to nude mice. Since then, this patient-derived xenograft (PDX) model has been maintained by serial transplantation for over 15 years. The tumour tissue was collected from the patient during surgery in 2001. Formal ethical approval by an ethics committee was optional according to Swedish law at that time and was regarded not necessary in this case. However, according to standard procedure, the patient provided oral informed consent for the collection which was verified by two surgeons, one pathologist and one medical physicist. The principles expressed in the Declaration of Helsinki were followed.

For this study, 4–5 weeks old female BALB/c nude mice (Charles River Laboratories, Sulzfeld, Germany and Janvier Labs, Le Genest-Saint-Isle, France) were subcutaneously transplanted in the neck with small GOT2 tumour tissue samples (*ca*. 1x1x1 mm$^3$) by trained research staff. Before transplantation, the mice were anesthetised by intraperitoneal (*i.p.*) injection of Ketaminol® vet. (Intervet AB, Stockholm, Sweden) and Domitor® vet. (Orion Pharma AB Animal Health, Sollentuna, Sweden), and an *i.p.* injection of Antisedan® vet. (Orion Pharma AB Animal Health) was used as antidote to anaesthesia. About 2 months later, small tumours were visible. The shape of the tumours was assumed to be that of an ellipsoid, and by measurements of the tumour length, width, and height, using digital callipers, the tumour volume was estimated. When the tumours reached a volume of about 100–1000 mm$^3$ (mean = 452 mm$^3$, SD = 208 mm$^3$), the mice were assigned to different treatment schedules, as described below. All animals had free access to autoclaved food and water throughout the study.

## Treatment schedules

GOT2-bearing nude mice (n = 51) were separated into groups of 5–8 animals and received EBRT alone, TKI alone, a combination of both, or were mock-treated as control (Table 1). Thus, four groups were used for studying the potentially increased effect by combining EBRT and vandetanib, and another four groups for EBRT and cabozantinib. The absorbed dose and amount of TKI was chosen to generate a low to moderate treatment effect as monotherapy to be able to detect any increased therapeutic effects from combination therapy. No animals showed any signs of treatment-related side effects.

EBRT was given as a single treatment on day 0 using a Varian linear accelerator with 6 MV nominal photon energy at 0-degrees gantry angle (Varian Medical Systems, Palo Alto, California, USA). Prior to irradiation, each mouse was anesthetised (Ketaminol® vet. and Domitor® vet.) and placed on its side (to avoid unnecessary irradiation of normal tissues) on a block of polystyrene. The centre of the tumour was placed at isocenter and tissue-equivalent material was placed around the mouse to obtain a relatively uniform dose distribution. In addition, 15 mm tissue-equivalent material was added on top of the tumour to place the centre of the tumour at a depth of about 20 mm (depending on the size of the tumour). Each tumour was individually irradiated with 3 Gy using a 30x30 mm$^2$ irradiation field.

TKI treatment was given as oral administration twice weekly starting on day 0. Vandetanib (Active Biochem LTD, Hong Kong, China) was obtained as a powder and dissolved in a solution of phosphate-buffered saline (PBS, Thermo Fisher Scientific, Waltham, Massachusetts, USA) and 1% polysorbate 80 (TWEEN® 80, Sigma-Aldrich, Saint Louis, Missouri, USA) and administered at a dose of 100 mg/kg. Cabozantinib (Active Biochem LTD, Hong Kong, China) was dissolved in sterile water and 50 mg/kg was administered. Prior to TKI administration, each mouse was weighed and the administered volume (0.15–0.20 ml) was adjusted to

**Table 1. Treatment schedules.** BALB/c nude mice carrying patient-derived medullary thyroid cancer (GOT2) were treated with the presented amounts and schedules of external beam radiotherapy (EBRT) and/or tyrosine kinase inhibitors (TKIs) vandetanib (Vand) or cabozantinib (Cabo), or a mock-treatment vehicle (solvent for vandetanib/cabozantinib). The vehicle volume (0.15 ml) was chosen to be similar to the administered volume for TKI treatment (0.15–0.20 ml depending on body weight).

| Group | Administered amount | | | Schedule, starting on day 0 | | | | |
|---|---|---|---|---|---|---|---|---|
| | EBRT (Gy) | TKI (mg/kg) | Vehicle (ml) | EBRT | TKI | Vehicle | n | T (d) |
| EBRT | 3 | - | 0.15 | Single | - | 2/w | 6 | 52 |
| Vand | - | 100 | - | - | 2/w | - | 5 | 52 |
| EBRT + Vand | 3 | 100 | - | Single | 2/w | - | 7 | 52 |
| Control | - | - | 0.15 | - | - | 2/w | 8 | 31 |
| EBRT | 3 | - | 0.15 | Single | - | 2/w | 5 | 49 |
| Cabo | - | 50 | - | - | 2/w | - | 8 | 49 |
| EBRT + Cabo | 3 | 50 | - | Single | 2/w | - | 5 | 49 |
| Control | - | - | 0.15 | - | - | 2/w | 7 | 28 |

n, number of animals in each group; F, follow-up time.

individualise the administered amount. On the same day as TKI administration, the control and EBRT monotherapy groups received mock-treatment by oral administration of the solvent solution (0.15 ml) for respective TKI (*i.e.* PBS with 1% polysorbate 80 or sterile water, Table 1). Since the solvent solutions differed between the TKIs, separate control and EBRT monotherapy groups were used for each TKI experiment.

## Treatment follow-up

After first treatment, the animals were monitored and tumour volume was measured twice weekly for 28–52 days depending on treatment group (Table 1). These follow-up times were chosen to allow for the tumours to regrow after a potential initial treatment effect. However, if a mouse met one of the following criteria before this point-in-time, it was killed and the tumour was resected and fixed in formalin: 1) the tumour exceeded a volume corresponding to 10% of the total body weight, 2) the body weight decreased by more than 10% after treatment, or 3) a mouse showed any signs of poor general condition. All mice were killed by cardiac puncture under anaesthesia (Pentobarbitalnatrium vet., Apotek Produktion & Laboratorier AB, Huddinge, Sweden).

## Immunohistochemistry

The formalin-fixed tumours were dehydrated, embedded in paraffin, and sliced into sections of 4 μm, according to standard procedures. To verify the MTC origin of the tumours, sections from the mock-treated control group were stained with haematoxylin and eosin (H&E) for morphological examination, and by using antibodies for MTC markers chromogranin A (CgA, dilution 1:500; ab68271, Abcam, Cambridge, England), synaptophysin (Syn, dilution 1:25; ab16659, Abcam), and calcitonin (Ctn, dilution 1:1000; A0576, Dako, Glostrup, Denmark). Antibodies were incubated for 1 hour.

For the immunohistochemical (IHC) staining, tumour sections were collected on glass slides and then treated with EnVision™ FLEX Target Retrieval Solution (high pH) using a PT-Link (Dako). The staining was done in an Autostainer Link using EnVision™ FLEX (Dako) according to the manufacturer's instructions. Positive and negative controls were included in each run. A microscope (20x magnification, Eclipse E1000, Nikon Instruments, Amsterdam, Netherlands) equipped with a camera (ProgRes C7, Jenoptik, Jena, Germany) was used for imaging of the stained tumour sections.

## Data calculations and statistical analyses

The relative tumour volume (RTV), defined as the tumour volume at a given time divided by the tumour volume at day 0 (start of treatment), was individually calculated for each mouse and used for the statistical analyses. First, to analyse overall differences between all group means, one-way ANOVA using GraphPad Prism 7.04 (GraphPad Software, La Jolla, San Diego, California, USA) was performed for each measurement day at which all groups were still followed (*i.e.* day 0–31 or day 0–28, Table 1). Statistically significant findings were further analysed in group-to-group comparisons using Student's t test. All reported p-values were adjusted using the Bonferroni-Holm method to counteract the problem of multiple comparisons [20]. For all tests, an adjusted p-value of less than 0.05 was considered statistically significant.

To analyse potential additive or synergistic effects, the Bliss independence model was used [21]. A predicted additive fractional response, at a given time, between EBRT and either of the two TKIs, $F_{add}$, was calculated:

$$F_{add} \; = \; F_{EBRT} + F_{TKI} - F_{EBRT}F_{TKI}, \qquad\qquad 1$$

where $F_{EBRT}$ and $F_{TKI}$ is the fractional response in an EBRT and a TKI monotherapy group at that time, respectively [22]. The fractional response in a monotherapy group, $F_i$, was calculated based on the RTVs using

$$F_i \; = \; 1 - \frac{RTV_{mono}}{RTV_{control}}, \qquad\qquad 2$$

where $RTV_{mono}$ and $RTV_{control}$ is the RTV in a monotherapy group and in the mock-treated control group, respectively. Then, the measured effect in the combination therapy group was assumed to have been synergistic, additive or antagonistic if it was higher than, equal to or lower than $F_{add}$, respectively.

## Results

### Vandetanib combined with radiation gave additive effects on tumour volume reduction

Vandetanib monotherapy resulted in a clear effect on tumour volume (Fig 1A–1C). Ten days after first treatment, the smallest mean RTV of 0.98 was reached ($RTV/RTV_{control} = 0.51$, p<0.001). However, even though vandetanib was administered throughout the entire follow-up period, the tumours started to grow again, but with a slower rate than the mock-treated control tumours. EBRT monotherapy also resulted in a clear effect on tumour volume (Fig 1A–1C). The smallest mean RTV for this treatment was 0.76 ($RTV/RTV_{control} = 0.40$, p<0.001) at day 10. However, because EBRT was only given as a single treatment on day 0, the tumours started to regrow again, with a rate similar to that of the mock-treated control tumours.

The combination of EBRT and vandetanib led to an increased anti-tumour effect compared with monotherapy (Fig 1A–1C). For example, two weeks after first treatment, the mean RTV in the combination therapy group was 0.60 ($RTV/RTV_{control} = 0.27$, p<0.001). This RTV was significantly lower than the corresponding mean RTV values of 0.80 (p = 0.006) and 1.06 (p<0.001) for the groups that received EBRT and vandetanib monotherapy, respectively. Even though increased effects were seen, the measured effect from combination therapy did not appear to be larger than the predicted additive effects at most days of measurement (Fig 1A).

In addition to resulting in the largest reduction in tumour volume, combination treatment also resulted in the longest time to progression (TTP) (Fig 1D). An animal was regarded to

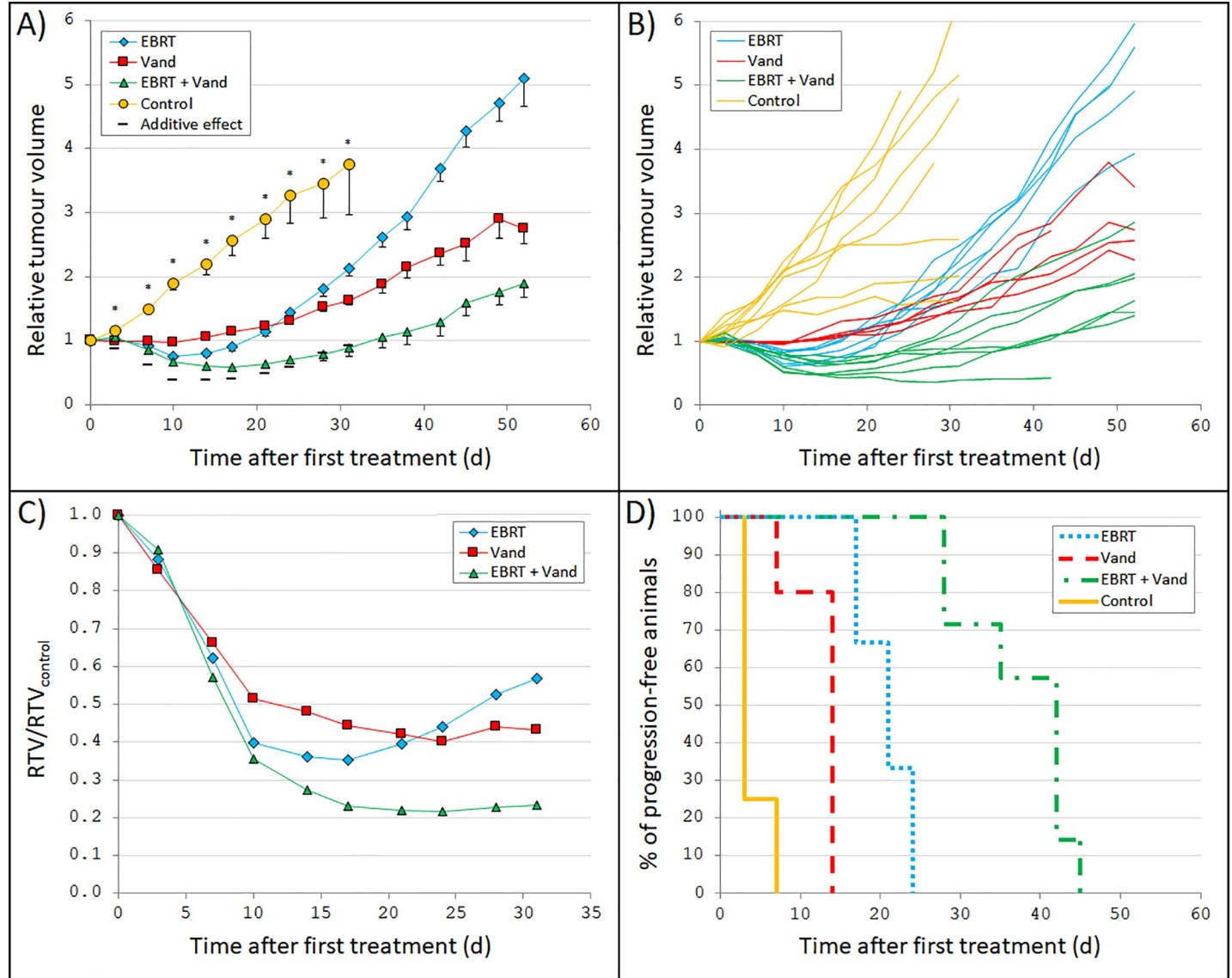

**Fig 1. Tumour growth after radiation and/or vandetanib treatment.** Nude mice carrying GOT2 tumours were treated with external beam radiotherapy (EBRT), vandetanib (Vand), a combination of both, or mock-treated as control. The absorbed dose from EBRT (3 Gy in one exposure on day 0) and the amount of vandetanib (100 mg/kg twice weekly) were chosen to give a moderate treatment effect to be able to detect any increased effect from combination therapy. **(A)** Mean relative tumour volume (RTV) *vs.* time after start of treatment. RTV was calculated as the tumour volume at a given time divided by the tumour volume at start of treatment. Included are also the corresponding calculated RTVs given a predicted additive effect (Eq. ((1)). Error bars represent SEM. The stars indicate that there was a statistically significant difference between the group means (ANOVA performed at day 3–31; $p < 0.05$). **(B)** Individual curves of RTV *vs.* time after start of treatment for each mouse. **(C)** Group-wise mean RTV compared with control (RTV/RTV$_{control}$) *vs.* time after start of treatment. Note the difference in x-axis scale. **(D)** Progression-free survival (percentage of animals without tumour progression) *vs.* time after start of treatment.

have reached tumour progression if RTV>1 (after initial volume reduction), or if the animal was killed. TTP was similar for the EBRT and vandetanib monotherapy groups. After about 20 days, 100% of the animals treated with EBRT or vandetanib monotherapy had reached tumour progression. In contrast, treatment with the combination of EBRT and vandetanib resulted in a much longer TTP, and tumour progression was not reached until 40 days after first treatment for 100% of the animals.

## Cabozantinib combined with radiation gave additive effects on tumour volume reduction

For cabozantinib monotherapy, a minimum mean RTV of 0.79 was reached ten days after first treatment ($RTV/RTV_{control} = 0.44$, $p < 0.001$) (Fig 2A–2C). After this minimum, a small increase in RTV with time was seen (much lower than for the control group). However, when the mice were killed about 50 days after start of treatment, the mean RTV in the cabozantinib

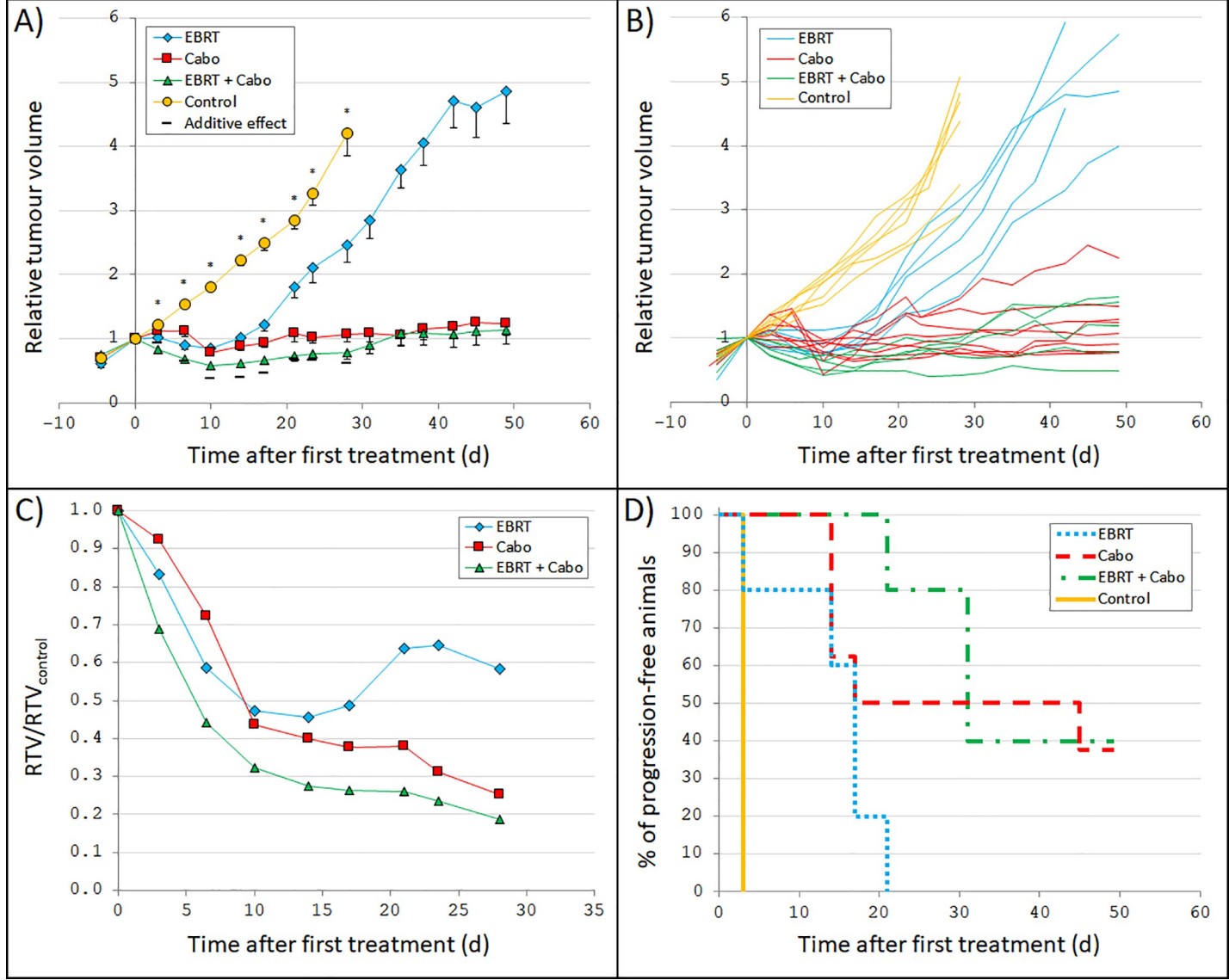

**Fig 2. Tumour growth after radiation and/or cabozantinib treatment.** Nude mice carrying GOT2 tumours were treated with external beam radiotherapy (EBRT), cabozantinib (Cabo), a combination of both, or mock-treated as control. The absorbed dose from EBRT (3 Gy in one exposure on day 0) and the amount of cabozantinib (50 mg/kg twice weekly) were chosen to give a moderate treatment effect to be able to detect any increased effect from combination therapy. **(A)** Mean relative tumour volume (RTV) *vs*. time after start of treatment. RTV was calculated as the tumour volume at a given time divided by the tumour volume at start of treatment. Included are also the corresponding calculated RTVs given a predicted additive effect (Eq. ((1)). Error bars represent SEM. The stars indicate that there was a statistically significant difference between the group means (ANOVA performed at day 3–28; $p < 0.05$). **(B)** Individual curves of RTV *vs*. time after start of treatment for each mouse. **(C)** Group-wise mean RTV compared with control ($RTV/RTV_{control}$) *vs*. time after start of treatment. Note the difference in x-axis scale. **(D)** Progression-free survival (percentage of animals without tumour progression) *vs*. time after start of treatment. Some lines end before they reach 0% because not all animals in these treatment groups had yet reached an RTV>1 when the groups were killed.

monotherapy group was still only 1.23. The animals that received EBRT monotherapy in the cabozantinib experiment also showed a clear effect on tumour volume with a minimum RTV of 0.86 (RTV/RTV$_{control}$ = 0.53, p<0.001) after ten days.

The combination of EBRT and cabozantinib resulted in an increased effect on tumour volume compared with monotherapy (Fig 2A–2C). For example, two weeks after start of treatment, the RTV for the combination therapy group was 0.61 (RTV/RTV$_{control}$ = 0.28, p<0.001), which was significantly lower than 1.02 (p = 0.004) and 0.89 (p = 0.026) for the EBRT and cabozantinib monotherapy groups, respectively. However, this measured increased effect did not appear to be larger than the predicted additive effects at most days of measurement (Fig 2A).

Cabozantinib monotherapy was administered throughout the entire follow-up period, and compared with EBRT monotherapy (single treatment on day 0), cabozantinib monotherapy resulted in a longer TTP (Fig 2D). Overall, the slowest regrowth was seen in the combination therapy group. For example, after about 20 days, 100% of the animals treated with EBRT monotherapy had reached tumour progression, compared with 50% of the animals treated with cabozantinib monotherapy, and only 20% of the animals in the combination therapy group.

## Immunohistochemistry verified specific MTC markers

All tumours stained for CgA, Syn and Ctn showed a high and specific expression of all three MTC markers. In addition, the morphology of the tumours was consistent with that of MTC (revealed by the H&E-stained sections), verifying the MTC origin of the GOT2 tumours. A representative example of one of the tumours is shown in Fig 3.

## Discussion

In the present study, we show that the anti-tumour effect on MTC from irradiation alone can be increased by co-treatment with TKIs vandetanib or cabozantinib. This increased effect appeared to be additive for both combination therapy schedules investigated. Ideally, a synergistic effect between the different treatments would be achieved. However, combination therapy can offer several potential benefits for patients also when an additive, or even antagonistic, effect is achieved [23]. Firstly, if two drugs with different toxicity profiles are used, the total amount of drug can be increased while still keeping the side effects for healthy organs at risk at an acceptable level. Secondly, since development of drug resistance is a common phenomenon for many cancer therapies, the option to switch treatment using another drug can be very valuable for some patients. Also, if two drugs are used for treatment at the same time, one of the drugs may inhibit development of resistance to the other, and vice-versa. Thirdly, for heterogeneous tumours, the use of multiple drugs can be crucial to be able to kill all cancer cells within the tumour if only part of the cells respond when one drug is used. The same argument can be made for multiple tumours within a patient, *e.g.* if the metastases differ from the primary tumour. Lastly, patient-to-patient variability is another reason to use combination therapy. Many drugs show different efficacy in different patients, and if multiple drugs are used, there is a higher probability that the patient will respond to at least one of the drugs [24]. It should be mentioned that the lack of synergistic effects in this study could be related to the fact that immunocompromised nude mice were used. Thus, radiation-induced immunomodulatory effects are probably limited and it is known that vandetanib and cabozantinib target immunomodulatory pathways, such as the vascular endothelial growth factor (VEGF) pathway [25, 26].

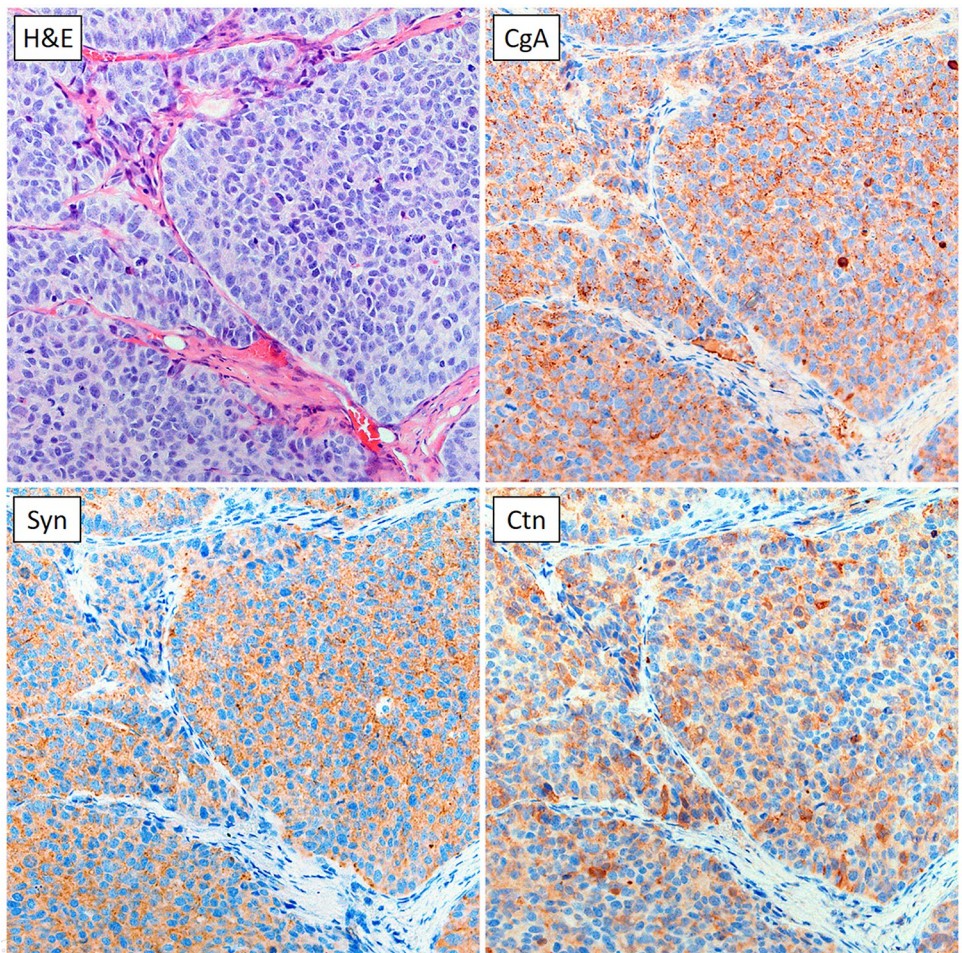

**Fig 3. IHC staining of a GOT2 tumour.** Tumours from mock-treated animals were harvested at the end of follow-up and stained with haematoxylin and eosin (H&E), and for three markers for MTC: chromogranin A (CgA), synaptophysin (Syn), and calcitonin (Ctn). In this representative example (at 20x magnification), a high and specific expression can be seen for all three markers.

As previously mentioned, most MTCs carry *RET* mutations. In addition, VEGF and its receptors are often overexpressed in MTC [27]. VEGF is a signal protein that stimulates angio-genesis, and hence tumour growth and metastasis formation. Therefore, drugs that target VEGF receptors should result in an anti-tumour effect. Vandetanib and cabozantinib are two TKIs that target both *RET* and VEGF receptors [25, 26, 28, 29]. As previously mentioned, these TKIs are both approved (by FDA and EMA) and recommended for first-line therapy in patients with advanced progressive MTC. In a large phase III trial, vandetanib showed an objective response rate of 45% and resulted in a median progression-free survival of 30.5 months compared with 19.3 months for placebo [9]. Also cabozantinib has been evaluated in a large phase III trial in which treatment resulted in a median progression-free survival of 11.2 moths versus 4.0 months for placebo and an objective response rate of 28% [10]. Final results on overall survival are not yet available. It should be noted that one of the inclusion criteria in the cabozantinib trial was that the patients were required to have a documented disease pro-gression, which could explain the longer progression-free survivals and higher objective response rate reported in the vandetanib trial. Nevertheless, the impact on progression-free

survival in both these phase III trials are very encouraging and TKIs offer a new treatment option for patients with metastatic MTC. Unfortunately, there are two major drawbacks of TKI treatment. Firstly, many patients experience significant treatment-related side effects, often severe enough to result in dose reduction or treatment discontinuation. These side effects are mainly associated with the gastrointestinal system (*e.g.* diarrhea, weight loss and decreased appetite, and nausea), but also rash and swelling is common [9, 10]. For example, dose reduction was required for 35% and 79% of the patients in the vandetanib and cabozantinib trials, respectively. Secondly, a well-known but unresolved issue for TKI treatment is that virtually all patients develop drug resistance with time. In the view of these drawbacks, combination therapy could be very beneficial for patients treated with TKIs as it could, at least in part, offer a solution to both of these problems, as previously discussed.

The combination of radiation treatment and TKIs for MTC could be especially interesting for PRRT (*e.g.* $^{177}$Lu-octreotate). Due to the lack of MTC animal models with an uptake of $^{177}$Lu-octreotate sufficiently high to result in large enough effect on tumour volume needed for this type of study, EBRT was used as radiation therapy instead of $^{177}$Lu-octreotate [30]. The tumour-to-blood (T/B) activity concentration ratio of GOT2-bearing mice is about 50 [30], while some MTC patients have a much higher expression of SSTRs and consequently higher T/B values (up to 350 for $^{111}$In-octreotide) and can therefore benefit from PRRT [12, 31]. Clinically, PRRT has been used to treat MTC in several studies [13–15, 32–34]. The results differed between studies, but generally, treatment was very well-tolerated and showed response rates of about 40–70% with an estimated prolonged median survival of about 1–2 years. In contrast to vandetanib and cabozantinib, PRRT with $^{177}$Lu-octreotate is not approved by FDA or EMA for treatment of MTC. However, $^{177}$Lu-octreotate was recently approved for another type of NET (specifically GEP-NETs) following the results of the first randomised controlled phase III trial for $^{177}$Lu-octreotate [17]. $^{177}$Lu-octreotate is usually administered according to a standardised protocol comprising 4–6 treatment cycles of 3.7 or 7.4 GBq. The total amount of activity administered is restricted by an absorbed dose to the kidneys of 23–28 Gy or 2 Gy to the bone marrow, whichever comes first. Haematological toxicity and late renal toxicity are the main problems for $^{177}$Lu-octreotate therapy [35, 36]. However, the rate of persistent haematological dysfunction is low and it is possible that the total amount of administered $^{177}$Lu-octreotate could be increased. This could result in an increased anti-tumour effect, while still keeping side effects at an acceptable level. Additionally, given the results in the present study, there is good reason to believe that the anti-tumour effect on MTC can be further increased by co-administration of vandetanib or cabozantinib.

In this experimental study, TKI monotherapy initially resulted in a clear effect on tumour volume. However, the tumours in the vandetanib group continued to grow again about 10 days after start of treatment, while the effect of cabozantinib treatment was more or less persistent throughout the entire follow-up. Although the dose-response relationship for each drug was not evaluated, this suggests that cabozantinib could be the choice of drug to treat GOT2 tumours. Clinically, the mean duration of response is similar for both drugs (about 15–20 months), and the question still remains whether vandetanib or cabozantinib should be used as first-line therapy for patients with metastatic MTC [1]. As previously mentioned, both drugs target *RET* and VEGF receptors, but there are additional targets that differ between the two drugs, namely epidermal growth factor (EGF) receptors for vandetanib, and MET (hepatocyte growth factor receptor) for cabozantinib. This difference could affect the choice of drug for individual patients. In the present study, also radiation monotherapy resulted in tumour regrowth (after initial treatment response), and after about 20 days, the growth rate appeared to be similar to that in the control groups. This could be explained by the fact that radiation therapy was only given as a single treatment on day 0, and repeated treatments would most

likely result in a maintained effect on tumour volume. As previously mentioned, this repeated treatment design is applied clinically for PRRT. If PRRT should be used in combination with TKI treatment, the fractionation schedule should be based on optimal synchronisation between these treatments.

The absorbed dose and the administered amounts of TKIs were chosen to give a low to moderate treatment effect as monotherapy to be able to detect any increased effects from combination therapy. Therefore, it is likely that the anti-tumour effects seen here could be significantly increased by higher treatment doses. The absorbed dose from EBRT was determined based on previous data in our research group, where 5 Gy delivered to GOT2 tumours resulted in an RTV of 0.57 at about two weeks after treatment [37]. The absorbed dose of 3 Gy used in this study instead resulted in a corresponding RTV of about 0.9, which was similar to the effect from TKI monotherapy (RTV≈1.0). The amount of cabozantinib was chosen as half of the vandetanib dose based on the relationship between the doses administered to MTC patients (300 and 140 mg/day for vandetanib and cabozantinib, respectively). Given the results for cabozantinib monotherapy in this study, a slightly lower dose could have been preferable. Also, it should be mentioned that the FDA-approved starting dose for cabozantinib of 140 mg/d have been questioned, and that a lower dose of 40–100 mg/d has been recommended in several clinical trials for prostate cancer after the approval of cabozantinib for MTC [38, 39].

## Conclusions

The results show that an increased therapeutic effect could be achieved when combining irradiation with TKIs vandetanib or cabozantinib for treatment of MTC. It is likely that this increased effect would be achieved also if [177]Lu-octreotate treatment, instead of EBRT, is used in combination with TKIs. The potential of combining [177]Lu-octreotate with vandetanib or cabozantinib for treatment of patients with MTC should be evaluated in future studies.

## Acknowledgments

The authors thank Gülay Altiparmak and Ulric Pedersen for their expert technical assistance with the IHC staining and image processing.

## Author Contributions

**Conceptualization:** Viktor Sandblom, Johan Spetz, Emman Shubbar, Mikael Montelius, Ola Nilsson, Eva Forssell-Aronsson.

**Data curation:** Viktor Sandblom, Emman Shubbar.

**Formal analysis:** Viktor Sandblom.

**Funding acquisition:** Viktor Sandblom, Eva Forssell-Aronsson.

**Investigation:** Viktor Sandblom, Emman Shubbar, Ingun Ståhl, John Swanpalmer, Ola Nilsson.

**Methodology:** Viktor Sandblom, Johan Spetz, Emman Shubbar, Mikael Montelius, John Swanpalmer, Ola Nilsson, Eva Forssell-Aronsson.

**Project administration:** Viktor Sandblom.

**Resources:** Emman Shubbar.

**Software:** Viktor Sandblom.

**Supervision:** Ola Nilsson, Eva Forssell-Aronsson.

**Visualization:** Viktor Sandblom, Ola Nilsson.

**Writing – original draft:** Viktor Sandblom.

**Writing – review & editing:** Viktor Sandblom, Johan Spetz, Emman Shubbar, Mikael Montelius, Ingun Ståhl, John Swanpalmer, Ola Nilsson, Eva Forssell-Aronsson.

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
