## [Decision Letter · Decision Letter 0]

1 May 2020

PONE-D-20-00504

Increased therapeutic effect on medullary thyroid cancer using a combination of radiation and tyrosine kinase inhibitors

PLOS ONE

Dear Mr Sandblom,

Thank you for submitting your manuscript to PLOS ONE. After careful consideration, we feel that it has merit but does not fully meet PLOS ONE’s publication criteria as it currently stands. Therefore, we invite you to submit a revised version of the manuscript that addresses the points raised during the review process.

Please be sure to address all comments raised by both Reviewers. Specifically regarding points #2 and #3 raised by Reviewer#2, please make your best effort to accommodate these recommendations. If there are circumstances that prevent full compliance with points#2 and #3, please explain how any partial compliance addresses the overall concern of each point, and why complete compliance is not feasible. 

We would appreciate receiving your revised manuscript by Jun 15 2020 11:59PM. To enhance the reproducibility of your results, we recommend that if applicable you deposit your laboratory protocols in protocols.io, where a protocol can be assigned its own identifier (DOI) such that it can be cited independently in the future. For instructions see: http://journals.plos.org/plosone/s/submission-guidelines#loc-laboratory-protocols

We look forward to receiving your revised manuscript.

Kind regards,

Donald P. Bottaro

Academic Editor

PLOS ONE

Journal Requirements:

2. We noticed you have some minor occurrence(s) of overlapping text with the following previous publication(s), which needs to be addressed:

https://doi.org/10.1371/journal.pone.0225260

In your revision ensure you cite all your sources (including your own works), and quote or rephrase any duplicated text outside the Methods section. Further consideration is dependent on these concerns being addressed.

Reviewers' comments:

Reviewer's Responses to Questions

**Comments to the Author**

1. Is the manuscript technically sound, and do the data support the conclusions?

Reviewer #1: Yes

Reviewer #2: Yes

2. Has the statistical analysis been performed appropriately and rigorously? 

Reviewer #1: Yes

Reviewer #2: Yes

3. Have the authors made all data underlying the findings in their manuscript fully available?

Reviewer #1: No

Reviewer #2: Yes

4. Is the manuscript presented in an intelligible fashion and written in standard English?

Reviewer #1: Yes

Reviewer #2: Yes

5. Review Comments to the Author

Reviewer #1: The authors used an orthotopic PDX model to test the efficacy of combining clinically approved TKI therapies (vandetanib or cabozantinib) with external beam radiation therapy (EBRT). They found an additive effect by such combinations. The data were appropriately analyzed and are presented fairly.

The authors appropriately discussed that EBRT is different from radiolabelled somatostatin analogues. Because of this, I would remove the mention of radiolabelled somatostatin analogues from the abstract and the introduction sections as they are misleading: I expected that they would use a model of radiolabelled somatostatin analogues. They can replace this with the more broad term "radiotherapy".

Because the authors used immunocompromised nude mouse models, the potential immunomodulatory effects of EBRT were not captured. This is important as both vandetanib and cabozantinib target the immunomodulatory VEGF pathway and cabozantinib targets other important immunomodulatory pathways such as Axl. This should be mentioned in the discussion section.

Reviewer #2: Comments to Authors:

In this manuscript, the authors studied the combination irradiation with tyrosine kinase inhibitors, vandetanib, and cabozantinib in a PDX model of MTC. The authors showed an additive inhibition of tumor growth in combination therapy. Patients with high expression SSTRs may benefit from PRRT in combination with tyrosine kinase inhibitors.

Specific comments:

1. What is the level of somatostatin receptors expression in GOT2 tumors? Please provide a reference.

2. The tumors should be stained for a marker of cell proliferation, cell death, or angiogenesis, to provide some insight into

the mechanism of an additive effect.

3. Figure 3. Provide quantification for all markers and show images in tumors treated with monotherapy and combination.

4. In the discussion, provide some insight into the mechanism by which this combination inhibits tumor growth.

5. The table 1 is hard to read. Please remove monotherapy and treatment in each row.

6. For immunohistochemistry, provide dilution of primary antibodies and the time of incubations.

7. Line 185: Please Describe the results of the study

8. Line 190: what is a clear effect?

9. Line 229: please revise the sentence for clarity.

10. Line 254: provide a title describing the data

11. Line 360: explain the treatment with fractionated 177Lu-octreotate

6. PLOS authors have the option to publish the peer review history of their article (what does this mean?). If published, this will include your full peer review and any attached files.

Reviewer #1: No

Reviewer #2: No

---

## [Author Response · Author response to Decision Letter 0]

8 May 2020

Journal Requirements

Authors’ comment: Postal codes were removed from the affiliations list. Figure captions and tables were moved to directly after the paragraph in which they were first cited.

2. We noticed you have some minor occurrence(s) of overlapping text with the following previous publication(s), which needs to be addressed: https://doi.org/10.1371/journal.pone.0225260

In your revision ensure you cite all your sources (including your own works), and quote or rephrase any duplicated text outside the Methods section. Further consideration is dependent on these concerns being addressed.

Authors’ comment: Rephrasing was made (outside the Methods section) at lines 70-76, 207-213, 255-260, 290-291, and 295-297.

Authors’ comment: The phrase that refers to these data was removed, lines 390-393. This data was not important for the discussion.

Editor Comments

Thank you for submitting your manuscript to PLOS ONE. After careful consideration, we feel that it has merit but does not fully meet PLOS ONE’s publication criteria as it currently stands. Therefore, we invite you to submit a revised version of the manuscript that addresses the points raised during the review process. Please be sure to address all comments raised by both Reviewers. Specifically regarding points #2 and #3 raised by Reviewer#2, please make your best effort to accommodate these recommendations. If there are circumstances that prevent full compliance with points#2 and #3, please explain how any partial compliance addresses the overall concern of each point, and why complete compliance is not feasible.

Review Comments to the Author

Reviewer #1

The authors used an orthotopic PDX model to test the efficacy of combining clinically approved TKI therapies (vandetanib or cabozantinib) with external beam radiation therapy (EBRT). They found an additive effect by such combinations. The data were appropriately analyzed and are presented fairly.

Comments

The authors appropriately discussed that EBRT is different from radiolabelled somatostatin analogues. Because of this, I would remove the mention of radiolabelled somatostatin analogues from the abstract and the introduction sections as they are misleading: I expected that they would use a model of radiolabeled somatostatin analogues. They can replace this with the more broad term "radiotherapy".

Authors’ comment: We have now made clarifications in the abstract (lines 23-24) and the introduction (lines 81-84). The intended strategy to target the metastases is to use TKIs in combination with radiolabelled somatostatin analogues, but the animal model limits the possibility to test the hypothesis with 177Lu-octreotate, and external irradiation was needed. It should be noted that the GOT2 model is, as far as we know, the MTC model so far developed with the highest expression of somatostatin receptors (ref [30] in the manuscript: Dalmo J, et al. Oncology Reports. 2012;27(1):174-81). As the reviewer states, we have presented this situation in Discussion.

Because the authors used immunocompromised nude mouse models, the potential immunomodulatory effects of EBRT were not captured. This is important as both vandetanib and cabozantinib target the immunomodulatory VEGF pathway and cabozantinib targets other important immunomodulatory pathways such as Axl. This should be mentioned in the discussion section.

Authors’ comment: The authors are thankful for this suggestion and have added text about this at the end of the first paragraph of the discussion (lines 316-319).

Reviewer #2

In this manuscript, the authors studied the combination irradiation with tyrosine kinase inhibitors, vandetanib, and cabozantinib in a PDX model of MTC. The authors showed an additive inhibition of tumor growth in combination therapy. Patients with high expression SSTRs may benefit from PRRT in combination with tyrosine kinase inhibitors.

Comments

1. What is the level of somatostatin receptors expression in GOT2 tumors? Please provide a reference.

Authors’ comment: There is no specific data on somatostatin receptor expression of GOT2 tumours. However, the biodistribution of 177Lu-octreotate in GOT2-bearing nude mice has been studied and information regarding tumour-to-blood activity concentration ratio was added (line 347-350). It should be noted that the GOT2 model is, as far as we know, the MTC model so far developed with the highest expression of somatostatin receptors, where a comparison is made with the TT animal model (ref [30] in the manuscript: Dalmo J, et al. Oncology Reports. 2012;27(1):174-81).

2. The tumors should be stained for a marker of cell proliferation, cell death, or angiogenesis, to provide some insight into the mechanism of an additive effect. 

3. Figure 3. Provide quantification for all markers and show images in tumors treated with monotherapy and combination.

Authors’ comments to #2 and #3: In the present study, the animals were followed for as long time as possible in order to study the long-term effects and determine progression-free survival. Thus, no tumour tissue was collected during the initial phase when the tumour volume was declining, which is the period when the mentioned parameters would be most interesting to study. Instead, the mice were followed until tumour regrowth, when the differences between groups probably would be minimal. Furthermore, the radiation dose was limited in order to not reduce the tumour volume too much and still be able to determine the tumour volume with reasonable accuracy.

However, the suggested experiments are interesting, but requires new animal studies to be performed, and would be interesting to include in future work trying to elucidate mechanisms. Such studies would, however, require at least six months to complete, and we would then like to include more molecular methods for an even better understanding of the mechanisms, thus generating data amounts too large to be able to present and discuss in the present paper.

4. In the discussion, provide some insight into the mechanism by which this combination inhibits tumor growth.

Authors’ comment: We have now included some more information on this issue of interest (line 316-319, and previously 372-375). As discussed, it is likely that the effect of the combination is not fully demonstrated due to the use of immunocompromised animals (which is necessary when using human tumour tissues).

5. The table 1 is hard to read. Please remove monotherapy and treatment in each row.

Authors’ comment: These suggested changes were made, together with some more, to make the table easier to read (Table 1).

6. For immunohistochemistry, provide dilution of primary antibodies and the time of incubations.

Authors’ comment: This information was added (line 160-162).

7. Line 185: Please Describe the results of the study

Authors’ comment: The meaning of the comment is unclear to us. However, we have interpreted it as that we should use a more descriptive subtitle, and thus included one (lines 194-195). Although not suggested by the reviewer, we have changed the following subtitle in a similar way (lines 242-243).

8. Line 190: what is a clear effect?

Authors’ comment: We have used this way of writing at several occasions when we have a statistically significant difference between the groups. The specific data are then presented in the previous or following sentence at such occasions. For clarification we also added a reference to Figure 1 at this line (line 200).

9. Line 229: please revise the sentence for clarity.

Authors’ comment: This sentence was revised (line 263-264).

10. Line 254: provide a title describing the data

Authors’ comment: The title is now descriptive (line 289).

11. Line 360: explain the treatment with fractionated 177Lu-octreotate

Authors’ comment: The word fractionated is not necessary here and has been deleted. The sentence is now clarified (line 402).

---

## [Editor Report · Decision Letter 1]

12 May 2020

Increased therapeutic effect on medullary thyroid cancer using a combination of radiation and tyrosine kinase inhibitors

PONE-D-20-00504R1

Dear Dr. Sandblom,

We are pleased to inform you that your manuscript has been judged scientifically suitable for publication and will be formally accepted for publication once it complies with all outstanding technical requirements.

With kind regards,

Donald P. Bottaro

Academic Editor

PLOS ONE
---

## [Editor Report · Acceptance letter]

14 May 2020

PONE-D-20-00504R1 

Increased therapeutic effect on medullary thyroid cancer using a combination of radiation and tyrosine kinase inhibitors 

Dear Dr. Sandblom:

I am pleased to inform you that your manuscript has been deemed suitable for publication in PLOS ONE. Congratulations! Your manuscript is now with our production department. 

With kind regards,

on behalf of

Dr. Donald P. Bottaro 

Academic Editor

PLOS ONE